# Dynamics of Humoral Immunity to Myxoma and Rabbit Hemorrhagic Disease Viruses in Wild European Rabbits Assessed by Longitudinal Semiquantitative Serology

Joana Coelho,[a,b] Henrique Pacheco,[a,b] Marta Rafael,[b,c,d] Saúl Jiménez-Ruiz,[b,c,d,e] Paulo Célio Alves,[b,c,f,g] Nuno Santos[b,c,g]

[a]CIISA—Centro de Investigação Interdisciplinar em Sanidade Animal, Faculty of Veterinary Medicine, University of Lisbon, Lisbon, Portugal

[b]CIBIO, Centro de Investigação em Biodiversidade e Recursos Genéticos, InBIO Laboratório Associado, Universidade do Porto, Vairão, Portugal

[c]BIOPOLIS Program in Genomics, Biodiversity and Land Planning, CIBIO, Vairão, Portugal

[d]SABIO-IREC, Research Group in Health and Biotechnology, Institute for Game and Wildlife Research, University of Castilla-La Mancha, Castilla-La Mancha, Spain

[e]GISAZ-ENZOEM, Animal Health and Zoonoses Research Group, Competitive Research Unit on Zoonoses and Emerging Diseases, University of Cordoba, Cordoba, Spain

[f]Departamento de Biologia, Faculdade de Ciências, Universidade do Porto, Porto, Portugal

[g]Estação Biológica de Mértola (EBM), CIBIO, Mértola, Portugal

**ABSTRACT**  Myxoma virus (MYXV) and rabbit hemorrhagic disease virus (RHDV) are important drivers of the population decline of the European rabbit, an endangered keystone species. Both viruses elicit strong immune responses, but the long-term dynamics of humoral immunity are imperfectly known. This study aimed to assess the determinants of the long-term dynamics of antibodies to each virus based on a longitudinal capture-mark-recapture of wild European rabbits and semiquantitative serological data of MYXV and RHDV GI.2-specific IgG. The study included 611 indirect enzyme-linked immunosorbent assay (iELISA) normalized absorbance ratios for each MYXV and RHDV GI.2 from 505 rabbits from 2018 to 2022. Normalized absorbance ratios were analyzed using log-linear mixed models, showing a significant positive relationship with the time since the first capture of individual rabbits, with monthly increases of 4.1% for antibodies against MYXV and 2.0% against RHDV GI.2. Individual serological histories showed fluctuations over time, suggesting that reinfections boosted the immune response and likely resulted in lifelong immunity. Normalized absorbance ratios significantly increased with the seroprevalence in the population, probably because of recent outbreaks, and with body weight, highlighting the role of MYXV and RHDV GI.2 in determining survival to adulthood. Juvenile rabbits seropositive for both viruses were found, and the dynamics of RHDV GI.2 normalized absorbance ratios suggest the presence of maternal immunity up to 2 months of age. Semiquantitative longitudinal serological data provide epidemiological information, otherwise lost when considering only qualitative data, and support a lifelong acquired humoral immunity to RHDV GI.2 and MYXV upon natural infection.

**IMPORTANCE**  This study addresses the long-term dynamics of humoral immunity to two major viral pathogens of the European rabbit, an endangered keystone species of major ecological relevance. Such studies are particularly challenging in free-ranging species, and a combination of longitudinal capture-mark-recapture and semiquantitative serology was used to address this question. Over 600 normalized absorbance ratios of iELISA, obtained from 505 individual rabbits in 7 populations over 5 years, were analyzed using linear mixed models. The results support a lifelong acquired humoral immunity to myxoma virus and rabbit hemorrhagic disease virus upon natural infection and suggest the presence of maternal immunity to the latter in wild juvenile rabbits. These results contribute to understanding the epidemiology of two viral diseases threatening this keystone species and assist in developing conservation programs.

Address correspondence to Nuno Santos, nuno.santos@cibio.up.pt.

The authors declare no conflict of interest.

**KEYWORDS** capture-mark-recapture, enzyme-linked immune serum assay, RHDV GI.2, antibody dynamics, *Oryctolagus cuniculus algirus*

The European wild rabbit (*Oryctolagus cuniculus*) is a keystone species across its native range, having a disproportionately large impact on the ecosystems relative to its biomass (1, 2). European rabbits originated in the Iberian Peninsula, where two genetically and morphologically dissimilar subspecies occur: *Oryctolagus cuniculus cuniculus*, which was later domesticated and spread globally by humans, and *Oryctolagus cuniculus algirus*, a wild subspecies endemic in southwestern Iberia (3, 4). The European rabbit is classified as endangered in its native range (5) due to population declines, with the subspecies *Oryctolagus cuniculus algirus* being peculiarly affected (6). Infectious diseases, notably myxomatosis and rabbit hemorrhagic disease, are considered major determinants of the decline of the European rabbit (5).

Myxoma virus (MYXV), a leporipoxvirus whose natural hosts are American *Sylvilagus* rabbits, causes myxomatosis in the European rabbit, a systemic disease with varying case fatality rates. This virus was introduced in France in 1952, spreading throughout Europe and causing population declines of >90% (7). The virus is currently endemic in the Iberian Peninsula (8). Rabbit hemorrhagic disease virus (RHDV) is a lagovirus whose variant GI.1 was first detected in 1984 in China (9) and subsequently identified in Europe in 1986. It causes a systemic disease, usually with a high case fatality rate (10, 11). In 2010, a new antigenically different variant, RHDV GI.2, was detected in France and subsequently spread throughout Europe, replacing RHDV GI.1 (12–15). This new variant again caused important declines in wild European rabbit populations (16).

European rabbits that survive myxomatosis or rabbit hemorrhagic disease develop strong humoral immunity (17, 18). While humoral immunity against both agents is considered lifelong, its long-term dynamics have seldom been assessed, particularly in wild populations. The probability of seroconversion (transition from seronegative to seropositive) was shown to be higher than the probability of seroreversion for MYXV and RHDV GI.1 in the European rabbit (19). Acquired immunity to MYXV and RHDV GI.1 has been shown to be transmitted to the offspring of seropositive does (20, 21), but evidence is lacking regarding RHDV GI. 2 (22), although it is expected to occur. Transmission of maternal antibodies against RHDV GI.2 has been reported in vaccinated domestic rabbits (23) but not in naturally infected wild ones.

Longitudinal serological studies, usually in a capture-mark-recapture design, play a key role in assessing long-term antibody dynamics (24, 25). Data obtained from tests that detect the presence of serum antibodies specific to the pathogen of interest consist of the absorbance of a sample, ideally normalized by the absorbance of a control (semiquantitative data). Such data are usually converted to binomial data (seropositive/seronegative) by comparison of the measured absorbance with a validated positivity threshold (26). The conversion of semiquantitative to binomial data leads to the loss of information that might otherwise provide valuable insight into the epidemiology of the pathogens of interest (24, 27).

Although MYXV and RHDV GI.2 are important drivers of the population of European rabbits, the long-term dynamics of specific antibodies are not entirely understood, particularly regarding RHDV GI.2 (22). Based on a longitudinal semiquantitative serological study, this study aims to describe the patterns and assess the factors influencing the dynamics of antibodies generated upon a natural infection with MYXV and RHDV GI.2 in wild European rabbits.

## RESULTS

Apparent seroprevalence varied across study sites but overall was 52.4% (95% confidence interval [$CI_{95}$], 48.4 to 56.3%) for MYXV and 39.1% ($CI_{95}$, 35.3 to 43.1%) for RHDV GI.2 (Table 1).

The MYXV log-linear mixed models (log-LMM) showed significant positive relationships between the log-MYXV normalized absorbance ratio (NAR) and the seroprevalence of MYXV,

**TABLE 1** Apparent seroprevalence for MYXV and RHDV GI.2 at each study site

| Population | Study site[a] | MYXV | | RHDV gl.2 | |
|---|---|---|---|---|---|
| | | Seroprevalence (%) | $CI_{95}$ (%) | Seroprevalence (%) | $CI_{95}$ (%) |
| Free-ranging | CLw | 66.3 | 55.4–75.7 | 31.3 | 22.2–42.1 |
| | MTw | 40.0 | 29.7–51.3 | 56.0 | 44.8–66.7 |
| | ALPw | 95.9 | 86.3–98.9 | 59.2 | 45.2–71.8 |
| | VPw | 96.7 | 83.3–99.4 | 43.3 | 27.4–60.8 |
| Fenced | CLf | 50.0 | 28.0–72.0 | 43.8 | 23.1–66.8 |
| | PNNf$_1$ | 26.7 | 14.2–44.5 | 30.0 | 16.7–47.9 |
| | PNNf$_2$ | 33.5 | 27.5–40.1 | 37.3 | 31.0–43.9 |
| | PNNf$_3$ | 55.5 | 46.5–64.1 | 36.1 | 28.1–45.1 |

[a]Study sites: CLw, Companhia das Lezírias; MTw, Mértola; ALPw, Alpiarça; VPw, Vale Perditos; CLf, Companhia das Lezírias; PNNf$_1$ to PNNf$_3$: Parque Natureza Noudar sites 1 to 3.

rabbit density, serological status for MYXV, and its interaction with the months January, February, May, and September. Significant negative relationships with body weight and the interaction between the serological status for MYXV and months since first capture and rabbit density were also found (Table 2 and Fig. 1).

The RHDV log-LMM showed significant positive relationships between log-RHDV GI.2 NAR, the seroprevalence of RHDV GI.2, and the serological status for RHDV GI.2 (seropositive/seronegative). A significant negative cubic relationship with body weight was also found (Table 3 and Fig. 2).

Rabbits that were seropositive for MYXV and RHDV GI.2 presented body weights as low as 0.206 kg (Fig. 1B and 2B) or an estimated age of 23 to 24 days. The overall RHDV GI.2 NAR tended to decline until approximately 0.5 kg (63 to 64 days of age), rising again with increasing body weight (Fig. 2B). The NAR of seropositive rabbits increased with seroprevalence in the population at the time of trapping (Fig. 1C and 2C), and that of MYXV-seropositive rabbits decreased with increasing population density at the time of trapping (Fig. 1D). The NARs of rabbits seropositive for MYXV were significantly higher in January, February, May, and September than in the reference month of July (Table 2 and Fig. 1E).

The NARs of seropositive rabbits significantly increased monthly by 4.1% for MYXV and 2.0% for RHDV GI.2 (Tables 2 and 3; Fig. 1A and 2A). Individual histories of NARs for both viruses showed varied fluctuating patterns but generally consisted of relatively sharp increases interspersed with slow declines (Fig. 3).

## DISCUSSION

We report the results of a longitudinal semiquantitative serological study of European rabbits from the southwestern Iberian subspecies *Oryctolagus cuniculus algirus*. Semiquantitative NARs of virus-specific IgG increased significantly with time since the first capture (Tables 1 and 2; Fig. 1A and 2A). The individual serological histories showed varied patterns, with several seroconversions but no clear seroreversions detected. Once seropositive, the NARs tended to fluctuate, with sometimes sharp increases followed by relatively slow declines (Fig. 3). We hypothesize that increases in the NARs of seropositive rabbits between consecutive captures occur in response to reexposure to the virus but that declining NARs probably relate to antibody decay over time (for examples, see references 25 and 28). Together, these findings suggest that immunity upon natural infections by MYXV and RHDV GI.2 is likely lifelong in European rabbits, with reinfections playing a role in boosting antibody levels and maintaining long-term humoral immunity, as reported for other host pathogens (29, 30).

While the time span of serological data for individual rabbits in this study was relatively short (the maximum interval between the first and last capture of individual rabbits was 36 months), it should be noted that the life span of wild European rabbits is also short, mainly due to predation and diseases (for example, see reference 31; reviewed in reference 32). A life span of 7.6 years was recorded in Australia, where rabbit predators are less common than in the Iberian Peninsula (33). In an experimental

**TABLE 2** Summary of the log-LMM of NARs for MYXV[a]

| Variable | $\beta$ | SE ($\beta$) | CI$_{95}$ ($\beta$)[b] |
|---|---|---|---|
| Intercept | −0.503 | 0.092 | **−0.675, −0.334** |
| Time since the first capture (mo) | 0.040 | 0.011 | **0.019, 0.061** |
| Serological status for MYXV | | | |
| Seropositive | 1.833 | 0.074 | **1.688, 1.973** |
| Sex | | | |
| Males | −0.003 | 0.034 | −0.096, 0.036 |
| Mo of sampling | | | |
| January | −0.207 | 0.118 | −0.422, 0.018 |
| February | −0.200 | 0.123 | −0.437, 0.035 |
| March | 0.0006 | 0.165 | −0.326, 0.304 |
| April | −0.386 | 0.107 | **−0.592, −0.185** |
| May | −0.070 | 0.092 | −0.253, 0.093 |
| June | −0.002 | 0.078 | −0.146, 0.151 |
| August | −0.336 | 0.194 | −0.737, 0.020 |
| September | −0.568 | 0.090 | **−0.738, −0.393** |
| October | −0.296 | 0.214 | −0.714, 0.099 |
| December | −0.451 | 0.408 | −1.275, 0.312 |
| Seroprevalence for MYXV | 0.356 | 0.108 | **0.152, 0.558** |
| Rabbit density | 0.054 | 0.027 | **0.004, 0.105** |
| Body wt | −0.070 | 0.038 | **−0.152, −0.002** |
| Body wt$^2$ | −0.011 | 0.020 | −0.049, 0.026 |
| Body weight$^3$ | 0.024 | 0.013 | −0.0002, 0.051 |
| Serological status for MYXV $\times$ mo since the first capture | −0.031 | 0.012 | **−0.054, −0.007** |
| Serological status for MYXV $\times$ rabbit density | −0.108 | 0.038 | **−0.179, −0.033** |
| Serological status for MYXV $\times$ January | 0.748 | 0.177 | **0.410, 1.088** |
| Serological status for MYXV $\times$ February | 0.873 | 0.217 | **0.460, 1.291** |
| Serological status for MYXV $\times$ March | −0.099 | 0.197 | −0.473, 0.282 |
| Serological status for MYXV $\times$ April | 0.262 | 0.258 | −0.229, 0.756 |
| Serological status for MYXV $\times$ May | 0.371 | 0.123 | **0.149, 0.614** |
| Serological status for MYXV $\times$ June | 0.137 | 0.108 | −0.069, 0.345 |
| Serological status for MYXV $\times$ August | 0.304 | 0.228 | −0.126, 0.751 |
| Serological status for MYXV $\times$ September | 0.383 | 0.126 | **0.151, 0.629** |
| Serological status for MYXV $\times$ October | 0.022 | 0.223 | −0.399, 0.449 |
| Serological status for MYXV $\times$ December | 0.320 | 0.468 | −0.568, 1.241 |

[a]Reference classes for the categorical variables: "seronegative," "female," and "July." Random effects: "individual" (intercept of the variance ± standard deviation, 0.030 ± 0.173), "year" (0.007 ± 0.083), and "site" (0.002 ± 0.039). Conditional $R^2$ = 0.899, marginal $R^2$ = 0.870. SE, standard error.
[b]Significant relationships highlighted in bold.

field enclosure in Central Europe, the average life expectancy of female rabbits reaching adulthood was estimated at 2.6 years (34). The timescale of individual serological histories in this study seems to be representative of the life spans of wild European rabbits.

Many rabbits seroconverted to a high NAR, but some seroconversions achieved only a low NAR, subsequently fluctuating around the threshold for seropositivity (Fig. 3). Based on the bibliography (35–39), we hypothesize that the level of specific serum IgG, achieved after seroconversion to MYXV or RHDV GI.2, might be associated with the interaction between the doses of virus to which individual rabbits are exposed (inoculum) and the individual immune competence. The infective dose of RHDV GI.2 was shown to be <$10^4$ viral genomes, but >$10^7$ genome copies was required to cause mortality (40). This highlights a large span of infective doses ($10^4$ to $10^7$) that could infect and subsequently immunize rabbits while not causing mortality. The minimum level of humoral immunity that protects against mortality upon infection in European wild rabbits is still to be determined, but preliminary results (unpublished data) suggest that rabbits at the indirect enzyme-linked immunosorbent assay (iELISA) cutoff threshold at a NAR equal to 2.0 are fully protected from mortality during RHDV GI.2 outbreaks.

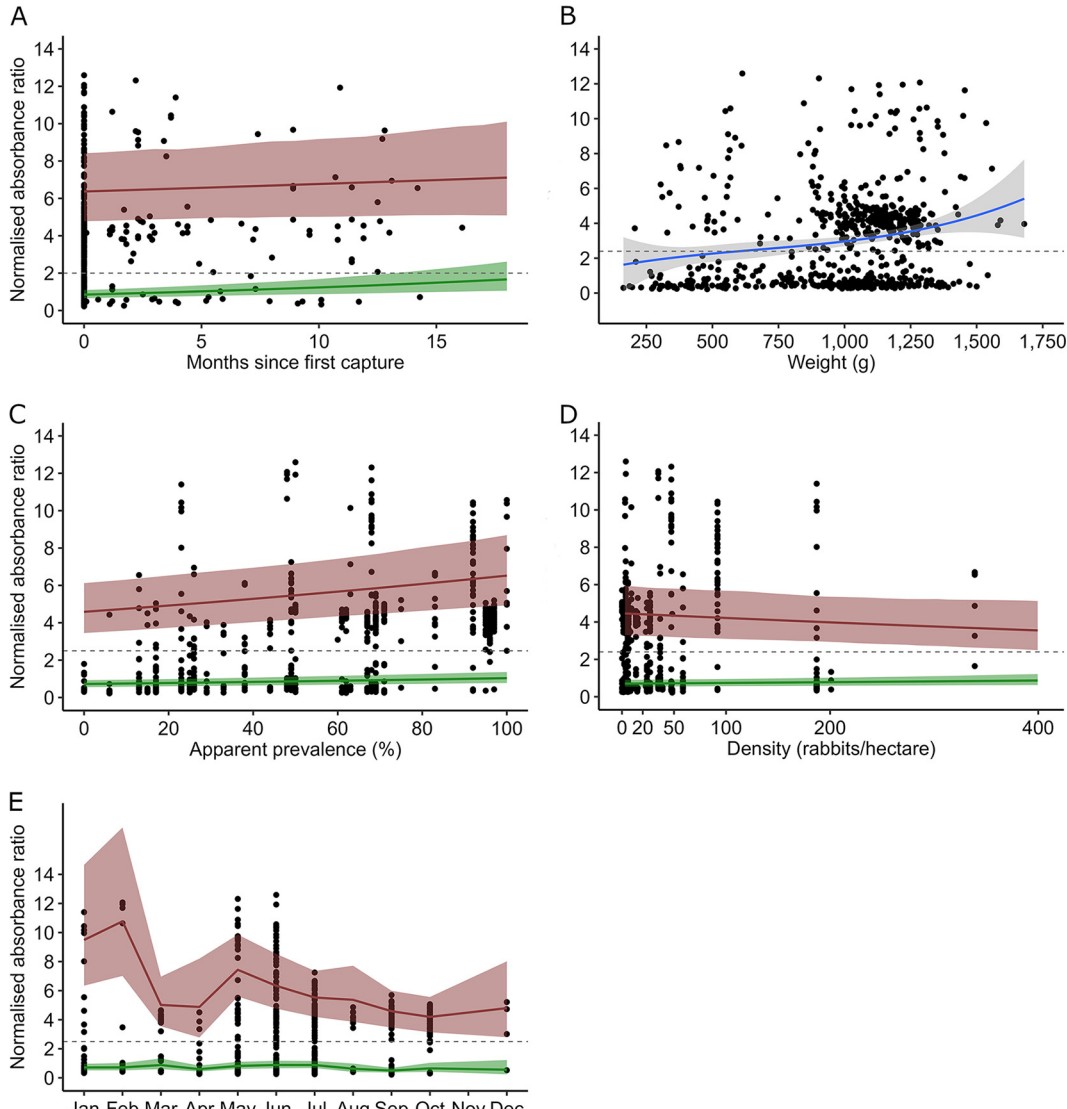

**FIG 1** Observed and predicted NARs for MYXV-specific IgG. NARs were observed (black dots) and predicted by the log-LMM for seropositive (red) and seronegative (green) rabbits. NARs are shown according to months since the first capture of each individual rabbit (A), body weight at the time of capture (cubic relationship) (B), apparent seroprevalence of MYXV in the population at the time of sampling (C), density of the rabbit population at the time of sampling (D), and month of capture (E). The cutoff threshold for seropositivity (NAR = 2.5) is represented by a horizontal dashed gray line.

Semiquantitative serology showed linear (MYXV) and cubic (RHDV) relationships with body weight, in which a higher NAR was detected in rabbits weighing >1 kg. Body weight was shown to have an almost linear relationship with age up to around 0.8 kg or 4 months (41). As rabbits weighing >1 kg are considered adults, our results suggest that survival to adulthood is associated with high NARs for both viruses, emphasizing their role in the population dynamics of European rabbits (16, 32). Nevertheless, many adult rabbits were seronegative for MYXV and RHDV GI.2, in contrast to what was described for *Oryctolagus cuniculus cuniculus* (31).

Rabbits that were seropositive for MYXV and RHDV GI.2 presented body weights starting from approximately 0.2 kg or 23 days of age, which could be explained by the presence of maternal antibodies or early infections (17). NARs of IgG specific to RHDV GI.2 tended to decline until about 0.5 kg or 63 to 64 days of age, suggesting the former hypothesis. Maternal immunity to RHDV GI.2 was shown to persist in juvenile rabbits up to 28 days of age in an experimental vaccination study (23). The role of passive transfer of maternal immunity to juvenile rabbits, resulting in herd immunity with a

**TABLE 3** Summary of the log-LMM of NARs for RHDV GI.2[a]

| Variable | $\beta$ | SE ($\beta$) | CI$_{95}$ ($\beta$)[b] |
|---|---|---|---|
| Intercept | −0.589 | 0.130 | **−0.831, −0.360** |
| Time since the first capture (mo) | 0.020 | 0.010 | **0.002, 0.041** |
| Serological status for RHDV | | | |
| Seropositive | 2.158 | 0.084 | **1.995, 2.319** |
| Sex | | | |
| Males | −0.030 | 0.038 | −0.101, 0.044 |
| Mo of sampling | | | |
| January | −0.076 | 0.126 | −0.395, 0.149 |
| February | 0.028 | 0.197 | −0.412, 0.345 |
| March | −0.344 | 0.185 | −0.692, 0.016 |
| April | −0.070 | 0.119 | −0.286, 0.179 |
| May | −0.108 | 0.083 | −0.264, 0.054 |
| June | −0.038 | 0.076 | −0.189, 0.105 |
| August | −0.130 | 0.144 | −0.429, 0.129 |
| September | −0.167 | 0.086 | −0.322, 0.029 |
| October | −0.228 | 0.141 | −0.585, 0.036 |
| December | 0.140 | 0.266 | −0.491, 0.608 |
| Seroprevalence for RHDV | 0.561 | 0.132 | **0.320, 0.807** |
| Rabbit density | 0.048 | 0.035 | −0.014, 0.119 |
| Body wt | 0.177 | 0.042 | **0.095, 0.255** |
| Body wt$^2$ | 0.010 | 0.021 | −0.033, 0.050 |
| Body wt$^3$ | −0.037 | 0.014 | **−0.064, −0.009** |
| Serological status for RHDV × mo since the first capture | −0.010 | 0.121 | −0.033, 0.014 |
| Serological status for RHDV × Rabbit density | −0.042 | 0.043 | −0.123, 0.042 |
| Serological status for RHDV × January | 0.307 | 0.228 | −0.133, 0.745 |
| Serological status for RHDV × February | −0.213 | 0.225 | −0.651, 0.215 |
| Serological status for RHDV × March | −0.041 | 0.212 | −0.451, 0.364 |
| Serological status for RHDV × April | −0.350 | 0.219 | −0.769, 0.077 |
| Serological status for RHDV × May | 0.076 | 0.135 | −0.209, 0.324 |
| Serological status for RHDV × June | 0.193 | 0.118 | −0.033, 0.419 |
| Serological status for RHDV × August | −0.219 | 0.256 | −0.717, 0.284 |
| Serological status for RHDV × September | −0.100 | 0.129 | −0.340, 0.157 |
| Serological status for RHDV × October | −0.047 | 0.142 | −0.311, 0.236 |
| Serological status for RHDV × December | −0.092 | 0.509 | −1.069, 0.892 |

[a]Reference classes for the categorical variables: "seronegative," "female," and "July." Random effects: "individual" (intercept of the variance ± standard deviation, 0.048 ± 0.220), "year" (0.028 ± 0.167), and "site" (0.034 ± 0.184). Conditional $R^2$ = 0.909, marginal $R^2$ = 0.839.
[b]Significant relationships highlighted in bold.

low case fatality rate, has been shown for MYXV (33, 42). Our results suggest that the same could occur with RHDV GI.2, although early infections of juvenile rabbits could also produce the observed pattern. Further analysis, including specific IgM to distinguish maternal immunity from recent infections, is necessary to fully address this hypothesis.

The seroprevalence across study sites for MYXV (52.4%; CI$_{95}$, 48.4 to 56.3%) was in accordance with results reported for Spain (8, 43, 44). In the log-LMM, the random effects "site" and "year" were associated with little variance in semiquantitative serological data, highlighting the endemic status of MYXV in wild populations. Annual outbreaks of MYXV occur predictably in every population in the Iberian Peninsula, in a stable endemic situation (8, 43, 44).

NARs for MYXV decreased with increasing rabbit density. In our longitudinal study, higher densities occur at the end of the breeding season, when the population is composed predominantly of juvenile rabbits, most of which are seronegative (Fig. 1B). On the other hand, the NARs for MYXV increased with seroprevalence in the population. High seroprevalence is indicative of recent outbreaks, which boost antibody levels in the surviving seropositive rabbits. Together, these results support a highly dynamic

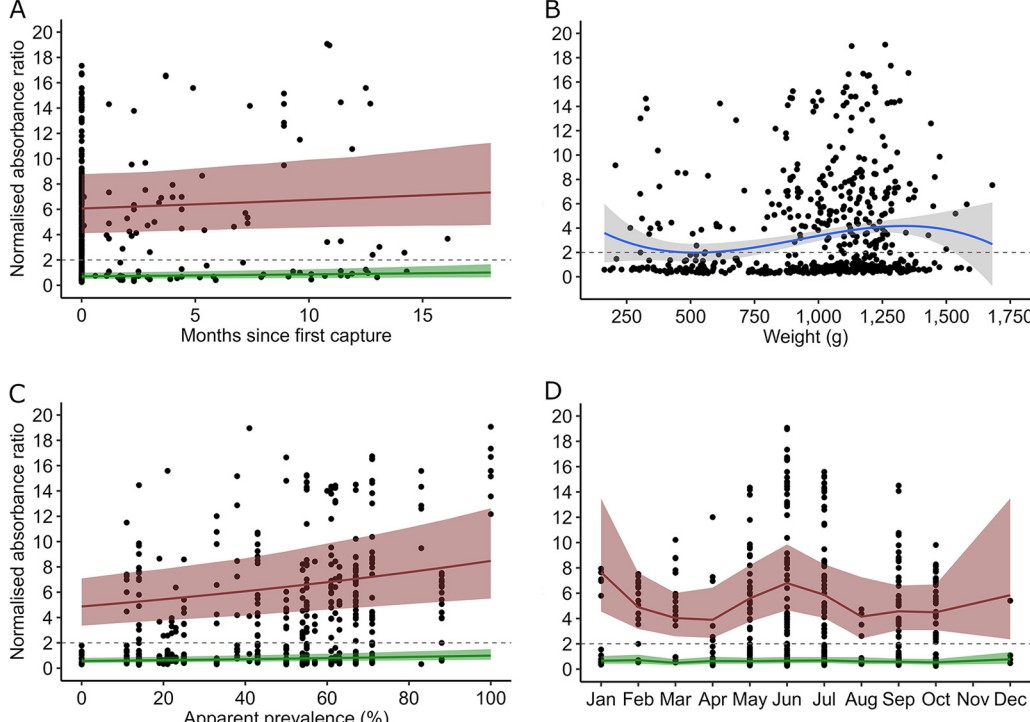

**FIG 2** Observed and predicted NARs for RHDV GI.2-specific IgG. NARs were observed (black dots) and predicted by the log-LMM for seropositive (red) and seronegative (green) rabbits. NARs are shown according to months since the first capture of each individual rabbit (A), body weight at the time of capture (cubic relationship) (B), apparent seroprevalence of RHDV in the population at the time of capture (C), and month of capture (D). The cutoff threshold for seropositivity (NAR = 2.0) is represented by a horizontal dashed gray line.

system in which outbreaks increase the overall antibody level in the population while breeding seasonally introduces many juvenile rabbits to the population, usually with low NARs (24). Such dynamics can explain the timing of the MYXV outbreaks, which tend to coincide or soon follow birth pulses (Fig. 1E) (21). Increased NARs, likely indicative of recent viral circulation, occur in December to February, May, and September (Fig. 1E), 2 to 3 months after the peak in rabbit births in the autumn and spring. These results support that MYXV outbreaks fade in rabbit populations once herd immunity is established and reemerge when the pool of susceptible hosts is reestablished by breeding pulses, as shown in this and other host-pathogen systems (21, 24, 45).

The seroprevalence across study sites for RHDV GI.2 (39.1%; CI$_{95}$, 35.3 to 43.1%) was significantly higher than that reported for wild rabbit populations in Portugal from 2013 to 2016 (26.5%; CI$_{95}$, 19.8 to 34.6% [46]). This observation suggests that herd immunity might be slowly building up in wild populations of the European rabbit in the Iberian Peninsula. We speculate that the persistence of this hypothetical trend will lead to Iberian rabbit populations achieving herd immunity (47) to RHDV GI.2. Long-term serological monitoring of wild rabbit populations is critical to test this prediction.

The random effects of "year" and "site" were associated with much variability in the semiquantitative iELISA results for RHDV GI.2. This observation highlights that RHDV GI.2, while endemic at large spatiotemporal scales, is characterized by localized outbreaks whose occurrence and extent varies considerably between years and populations (48). NARs for RHDV GI.2 increased with seroprevalence in the population, probably mediated by recent outbreaks of disease, but were not related to rabbit density (Table 3 and Fig. 2C).

**Conclusion.** This study highlights the importance of the longitudinal monitoring of wildlife diseases, coupled with semiquantitative serological data, to better understand the epidemiology and long-term dynamics of antibodies to myxoma and rabbit hemorrhagic disease GI.2 viruses in this endangered keystone species in Iberian Mediterranean ecosystems. Our results suggest that the humoral immunity to MYXV and RHDV GI.2 might be

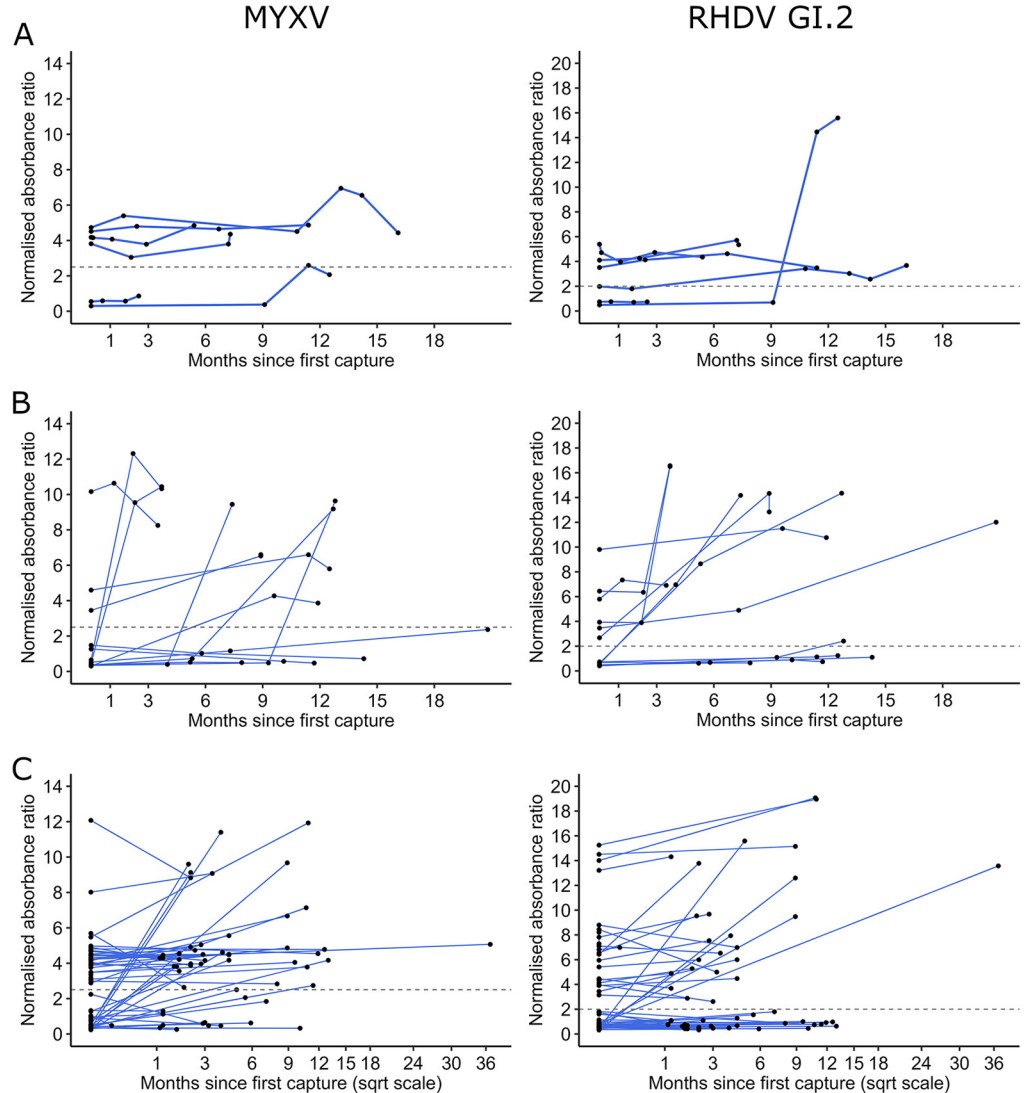

**FIG 3** Individual histories of NARs of MYXV- and RHDV-specific IgG. Individual histories are shown for ≥4 captures (A), 3 captures (B), or 2 captures (in square root scale) (C), with serological semiquantitative data for MYXV and RHDV GI.2. The cutoff threshold for seropositivity (NAR of 2.0 for RHDV and 2.5 for MYXV) is represented by a horizontal dashed gray line.

lifelong and support the role of reinfections in maintaining this acquired immunity in the face of antibody decay.

## MATERIALS AND METHODS

**Study design.** A longitudinal robust capture-mark-recapture study (49) of European rabbits from the southwestern Iberian subspecies *Oryctolagus cuniculus algirus* was performed at several locations in southern Portugal: two free-ranging populations at Companhia das Lezírias (CLw; 38°50′44′′N, 8° 51′49′′W) and Mértola (MTw; 37°43′27′′N, 7°40′34′′W) and fenced populations in four enclosures of 0.3 to 4.7 ha at Parque Natureza Noudar (PNNf$_1$ to PNNf$_3$; 38°11′04′′N, 7°02′24′′W) and Companhia das Lezírias (CLf; 38°50′34′′N, 8°48′30′′W). Captures were occasionally performed in two other free-ranging populations: Vale Perditos (VPw; 37°49′18′′N, 7°22′45′′W) and Alpiarça (ALPw; 39°15′25′′N, 8°33′26′′W).

The landscape at the study sites consists of a mosaic of scrub (mostly *Cistus* sp., *Lavandula* sp., and *Ulex* sp.) sparsely forested with cork oak (*Quercus suber*) at CLw and ALPw and holm oak (*Quercus ilex*) at PNNf, MTw, and VPw. The MTw and VPw populations were harvested. The populations at CLw and ALPw were not managed. At MTw and VPw, natural food is supplemented with cereal, while at the fenced sites (PNNf and CLf), water and commercial feed are provided *ad libitum* year-round, and predation by terrestrial carnivores is prevented by 2-m-high fences with perimeter electrical wire.

For the free-ranging populations, 30 to 52 cage traps were set regularly spaced in an area of approximately 13 ha/location. At the fenced sites,10 to 15 cage traps were placed near the feeders in each of the enclosures. Traps were set 2 h before sunset, baited with vegetables, and checked 2 h after sunset and again 1 h after sunrise, with them kept closed during the day.

At CLw, 19 sessions with 2 to 5 occasions (nights) each were performed between May 2018 and July 2021, during which 58 rabbits were captured 130 times. At MTw, 5 sessions with 2 to 3 occasions each were performed between April 2021 and June 2022, during which 124 rabbits were captured 185 times. At PNNf, 16 sessions with 1 to 3 occasions each were performed between July 2019 and June 2022, during which 453 rabbits were captured 651 times. At CLf, 4 sessions with 2 to 3 occasions each were performed between April 2021 and January 2022, during which 121 rabbits were captured 131 times. At VPw, 1 session with 2 occasions was performed in September 2020, during which 56 rabbits were captured. At ALPw, 2 sessions with 1 occasion were performed in September 2020 and 2021, during which 60 rabbits were captured.

**Sample and data collection.** Each rabbit was individually identified with a subcutaneous microchip when first captured. Up to 1.5 mL of whole blood (<0.25% body weight) was collected by venipuncture of the saphenous vein and placed in a clotting tube. The blood sample was then centrifuged at 1,430 × $g$ for 10 min, and the sera were stored at −20°C until serological analyses. Sex was assessed by visual inspection of the external genitalia, and weight was measured with scales (1-g precision). Weight correlates well with age up to 0.8 kg or 4 months of age, according to validated growth curves (41). Rabbits were released where captured immediately after processing. Live trapping and sample collection were conducted under permits ICNF 580/2018/CAPT, 8/2019/CAPT, 197/2020/CAPT, 23/2021/CAPT, and 2-DGVF/DRCA/2021 and according to European Union directives on the protection of animals used for scientific purposes (Directive 2010/63/EU) and international wildlife standards (50).

The population density at each sampling session was estimated using Jolly-Seber-Arnason-Schwarz models, either spatially explicit for the free-ranging sites or nonspatial for the fenced sites (51, 52). Both types of models were implemented using the package "openCR" (53) in R 3.6.1 (54).

**Serological assays.** All the employed serological assays were iELISAs. The in-house iELISA targeting RHDV GI.2-specific IgG was performed as described by Bárcena et al. (14) and Rouco et al. (46) with minor adaptations. Briefly, GI.2-derived virus-like particles (14), expressed in a baculovirus expression system and purified as previously described (55), were absorbed onto Nunc Maxisorp 96-well ELISA plates (100 ng/well) diluted in carbonate/bicarbonate buffer (pH 9.5) and incubated overnight at 4°C. The plates were blocked with phosphate-buffered saline (PBS)–0.05% Tween 20 in 5% skim milk solution and washed 3 times, and the sera were assayed at a 1/200 dilution in PBS–5% skim milk solution. Subsequently, the goat anti-rabbit IgG–horseradish peroxidase conjugate (Bio-Rad, Portugal) was added at a 1/4,000 dilution, followed by the addition of 3,3′,5,5′-tetramethylbenzidine (Abcam, UK). Reactions were stopped with 100 $\mu$L of 1 M phosphoric acid, and the optical density at 450 nm ($OD_{450}$) was recorded within 15 min. Positive controls consisted of pooled sera from rabbits with high iELISA readings (56), and negative controls consisted of pooled sera from unvaccinated domestic European rabbits without a history of clinical disease and kept in-house. The commercial iELISA kit (Ingezim 17.MIX.K1; Ingenasa, Spain) targeting MYXV-specific IgG was performed according to the manufacturer's instructions.

All serum samples and controls were tested in duplicate. The assays were considered valid if the average $OD_{450}$ of the two replicates of the positive control was >5 times the average $OD_{450}$ of the two replicates of the negative control. The assay results were standardized as normalized absorbance ratios (NARs) (57) according to the following equation:

$$NAR = \frac{\text{average OD450 sample}}{2 \times (\text{average OD450 negative control})}$$

NARs have no units, as they represent the ratio of the absorbance of the sample to that of the negative control. The serological status (seropositive/seronegative) of each rabbit was attributed based on the cutoff thresholds estimated by finite mixture models, being a NAR of 2.0 for RHDV GI.2 and a NAR of 2.5 for MYXV (56).

**Statistical analysis.** The data set consisted of 1,222 NARs, 611 each for MYXV and RHDV GI.2, from 505 individuals. Log-LMM with Gaussian error distribution were used to assess the effect on the NAR of the following independent variables: "year," "individual" rabbit, and "site" (random effects) and "month" of capture, "sex," and "serological status" for MYXV (in the MYXV model) and for RHDV GI.2 (in the RHDV model) (categorical fixed effects), apparent "seroprevalence" in the population at the time of sampling, "months since first capture," and rabbit population "density" (continuous fixed effects), and body "weight" (continuous cubic fixed effect). Reference classes for the categorical variables were, respectively, "July," "females," and "seronegative." The interaction between the "serological status" for MYXV (in the MYXV model) and RHDV GI.2 (in the RHDV model) and the variables "months since the first capture," "month" of capture, and rabbit "density" were also included in the models. Weight was included as a proxy of age, given that these variables are well correlated up to 800 g or 4 months of age (41).

Log-LMM were implemented using the package "lme4" (58) in R 3.6.1 (54). The variance inflation factor corrected for the number of degrees of freedom [$GVIF^{1/(2 \times df)}$] of each independent variable was estimated with a threshold for acceptance of 2.5. Continuous variables were standardized to their Z-scores. The marginal and conditional $R^2$ values of the models were estimated as described by Nakagawa and Schielzeth (59), implemented in the package "MuMIn" (60). The assumption of normality of the model residuals was checked by inspection of quantile-quantile plots. Graphics were produced using the package "ggplot2" (61). The predicted NARs were estimated from each model using the package "merTools" (62). The function "predictInterval" was used, which fits multivariate normal distributions to the random and fixed effects. One thousand values were sampled from these distributions for each category of the random and fixed effects, capturing the full uncertainty in predictions as 95% $CI_{95}$s.

**Data availability.** All data used in this study can be found at the Dryad repository: https://doi.org/10.5061/dryad.t1g1jwt74.

## ACKNOWLEDGMENTS

We acknowledge the support of Companhia das Lezírias S.A. and Empresa de Desenvolvimento e Infra-estruturas do Alqueva S.A./Parque de Natureza de Noudar, as well as the volunteers who assisted with the fieldwork.

We have no conflicts of interest to declare.

This work was supported by Fundação para a Ciência e Tecnologia (grant SFRH/BPD/116596/2016 to N.S.) and cofunded by project NORTE-01-0246-FEDER-000063, supported by Norte Portugal Regional Operational Program (NORTE2020), under the PORTUGAL 2020 Partnership Agreement, through the European Regional Development Fund (ERDF). S.J-R. was supported by postdoctoral contract Margarita Salas (University of Castilla-La Mancha) from the Program of Requalification of the Spanish University System (Spanish Ministry of Universities) financed by the European Union-NextGenerationEU and a research mobility grant within the Plan Propio of the University of Castilla-La Mancha for the year 2022.

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
