## [Reviewer comments · Microbiology Spectrum]

Microbiology Spectrum

Dynamics of humoral immunity to myxoma and rabbit haemorrhagic disease viruses in wild European rabbits assessed by longitudinal semi-quantitative serology

Joana Coelho, Henrique Pacheco, Marta Rafael, Saúl Jiménez-Ruiz, Paulo Alves, and Nuno Santos

Corresponding Author(s): Nuno Santos, Associacao BIOPOLIS - Rede de Investigacao em Biodiversidade e Biologia Evolutiva

Review Timeline:

Submission Date:	January 5, 2023
Editorial Decision:	February 20, 2023
Revision Received:	March 30, 2023
Accepted:	May 19, 2023

Editor: Daniela Rajao

Reviewer(s): Disclosure of reviewer identity is with reference to reviewer comments included in decision letter(s). The following individuals involved in review of your submission have agreed to reveal their identity: Paulina Niedzwiedzka-Rystwej (Reviewer #3)

Transaction Report:

DOI: <https://doi.org/10.1128/spectrum.00050-23>

February 20, 2023

Dr. Nuno Santos
Associacao BIOPOLIS - Rede de Investigacao em Biodiversidade e Biologia Evolutiva
CIBIO
Campus de Vairao
Vairao 4485-661
Portugal

Re: Spectrum00050-23 (Dynamics of humoral immunity to myxoma and rabbit haemorrhagic disease viruses in wild European rabbits assessed by longitudinal semi-quantitative serology)

Dear Dr. Nuno Santos:

Thank you for submitting your manuscript to Microbiology Spectrum. Your manuscript has now been reviewed by experts in the field. Although reviewers agreed that this manuscript provides interesting findings and valuable information, minor issues were pointed that need to be addressed, particularly improving the Language and clarity of the information.

Link Not Available

Sincerely,

Daniela Rajao

Journals Department
Reviewer comments:

Reviewer #1 (Comments for the Author):

This is an interesting manuscript that considers serological data indicating myxoma virus and rabbit haemorrhagic disease virus infections among wild rabbits of the *Oryctolagus cuniculus algirus* sub-species. Generally, it is well written, and the standard of English is good with few minor errors (e.g., in line 57 'later' should be replaced with 'latter'). Nonetheless, the English might be more straight-forward in places. For example, at line 129, 'Rabbits . . . presented body weights . . .' could be written 'Some rabbits . . . weighed as little as 0.206 kg, or about 23 - 24 days old.' A few sentences seem incomplete, for example at line 136

'The NAR . . . was significantly higher . . .' Than what?

Nonetheless, these are relatively minor issues.

The manuscript might be improved, and better aligned with the objectives of the journal which seeks 'research studies that are of high technical quality and are useful to the community' if there was less emphasis on blunt mathematical analysis and more attention was paid to sorting out epidemiological patterns or the ecology of the two diseases.

For example, although the authors state that evidence is lacking on whether maternal antibodies are passed on to young rabbits in the case of RHDV G1.2, it would be extremely surprising if it were not the case. This is a function of the rabbit immune system, not the virus. That could have been accepted as prior knowledge instead of it being 'confirmed' by the present study. Looking at the graphs presented there are far more interesting and useful conclusions that could have been drawn from the data with a more specific analysis.

For example, Fig 1 (E) and Fig 2 (D) imply that there are very different patterns of outbreaks of RHDV and MYXV during the year with RHDV possibly having a bimodal pattern and MYXV being most prevalent in winter. However, without a proper analysis that corrects for the age of rabbits in the sampled populations in a more useful way it is not possible to sort out whether this might be due to vectors (e.g., rabbit flea abundance reaching a peak in winter) or interactions between the two viruses. For example, if RHDV killed many young rabbits that had not yet been exposed to the myxoma virus, the seroprevalence of MYXV antibodies could increase simply because the population would then include mostly older rabbits that had previously contracted the disease.

In a second example, the authors note . . . that many young rabbits were seropositive for RHDV antibodies and assume these to be of maternal origin. The proportion of young rabbits with such antibodies declines until rabbits are about 63-64 days old then increases again, presumably because, on losing maternal antibody protection they have a higher chance of becoming infected with RHDV. However, there was no further analysis to support that evidence. Quite clearly, based on available data, only a proportion of the adult population was seropositive and showed evidence of prior exposure to RHDV, and so it would be expected that only a similar proportion of new-born young could show maternal antibodies. The broad statistical methods used do not appear to have been adequate to bring to the fore subtle inferences of this kind. Instead, careful sorting of the most relevant data (e.g., separating out seropositive young rabbits or seeking alternative ways of establishing the origin of antibodies) would be necessary. Maternal antibodies are IgG isotype alone and iso-ELISA tests could have been used to confirm their origin.

Figure 3 is confused as the authors seek different ways of presenting their evidence that rabbit antibodies are boosted by re-exposure or reinfection with the disease. As it has been accepted for over 20 years that rabbit antibodies increase sharply on re-exposure, it would have been more useful to consider whether seropositive rabbits showing evidence of an antibody boost were more commonly detected in different seasons or months of the year. Such information would provide an independent assessment of the timing of RHDV outbreaks and help to interpret earlier presented data on the seasonal changes in antibody prevalence.

In short, although the authors consider that 'longitudinal serological studies play a key role' in understanding disease, I feel that they could have applied better analyses to the data available by initiating the manuscript with some specific hypotheses to be tested. This would have helped to focus the statistical analyses and helped in reaching more useful, or at least less speculative conclusions.

Reviewer #2 (Comments for the Author):

The manuscript by Coelho et al presents the results of an extensive longitudinal field analysis on the development of protective anti viral immunity against important rabbit pathogens. The effort is remarkable and data presented indicates formation of lifelong protection after a few rounds of infection. While I liked the concept and the data, the presentation requires some rewording. In general, figures are hard to understand and pertinent information is not well conveyed. Please edit accordingly

Specific comments:

Figures: What is the unit of iELISA result? It is not indicated in the legend.

Figure 1B: What is the point of correlating with weight? Are these mice healthy and just bigger or overweight so obesity and immune system interplay is questioned?

Figure 3 is very hard to follow. Some other way if interpretation that includes how many times the rabbit is recaptured as a variable should be included.

Overall: Figure legends are too vague. More description is required for the readership to follow the complex data presented. For instance there is a dashed line in all graphs of but it is not acknowledged in any of the legends.

Discussion: "The role of the passive transfer of maternal immunity to juvenile rabbits, resulting in herd immunity with a low case-fatality rate, was shown for MYXV (33, 42). Our results suggest the same could occur with RHDV G1.2, but further analysis including specific IgM will be necessary to fully address this hypothesis." Why is IgM analysis required to test this hypothesis? IgM does not cross placenta. I do not get the relevance.

General: Did authors perform any studies on Ig subtypes? Do they predict any correlation with protection and certain isotypes?

Reviewer #3 (Comments for the Author):

This is an extremely valuable paper for the researchers interested in RHDV and Myxoma, showing how monitoring of the animals in nature may contribute to expanding the biological features of the disease and the characteristics of the virus. The team deals with this kind of animal trapping for several years, I have no doubts that the study has been conducted properly. The paper is written with a very high consciousness and knowledge about the subject. The results are properly documented. The conclusion is very crucial, showing the nature of the infection and the humoral response to the infection in the natural conditions for the first time, especially in the means of Lagovirus europaeus Gl.2. I have no concerns about the paper and in my opinion it is worth publishing.

Staff Comments:

Preparing Revision Guidelines

Please return the manuscript within 60 days; if you cannot complete the modification within this time period, please contact me. If you do not wish to modify the manuscript and prefer to submit it to another journal, please notify me of your decision immediately so that the manuscript may be formally withdrawn from consideration by Microbiology Spectrum.

The manuscript by Coelho et al presents the results of an extensive longitudinal field analysis on the development of protective anti viral immunity against important rabbit pathogens. The effort is remarkable and data presented indicates formation of lifelong protection after a few rounds of infection. While I liked the concept and the data, the presentation requires some rewording. In general, figures are hard to understand and pertinent information is not well conveyed. Please edit accordingly

Specific comments:

Figures: What is the unit of iELISA result? It is not indicated in the legend.

Figure 1B: What is the point of correlating with weight? Are these mice healthy and just bigger or overweight so obesity and immune system interplay is questioned?

Figure 3 is very hard to follow. Some other way if interpretation that includes how many times the rabbit is recaptured as a variable should be included.

Overall: Figure legends are too vague. More description is required for the readership to follow the complex data presented. For instance there is a dashed line in all graphs of but it is not acknowledged in any of the legends.

Discussion: "The role of the passive transfer of maternal immunity to juvenile rabbits, resulting in herd immunity with a low case-fatality rate, was shown for MYXV (33, 42). Our results suggest the same could occur with RHDV GI.2, but further analysis including specific IgM will be necessary to fully address this hypothesis." Why is IgM analysis required to test this hypothesis? IgM does not cross placenta. I do not get the relevance.

General: Did authors perform any studies on Ig subtypes? Do they predict any correlation with protection and certain isotypes?

Reply to the reviewers:

Reviewer #1 (Comments for the Author):

*This is an interesting manuscript that considers serological data indicating myxoma virus and rabbit haemorrhagic disease virus infections among wild rabbits of the *Oryctolagus cuniculus algirus* sub-species. Generally, it is well written, and the standard of English is good with few minor errors (e.g., in line 57 'later' should be replaced with 'latter'). Nonetheless, the English might be more straight-forward in places. For example, at line 129, 'Rabbits . . . presented body weights . . .' could be written 'Some rabbits . . . weighed as little as 0.206 kg, or about 23 - 24 days old.' A few sentences seem incomplete, for example at line 136 'The NAR . . . was significantly higher . . .' Than what?*

Reply: We acknowledge the comments and revised sentences throughout the manuscript.

Nonetheless, these are relatively minor issues.

The manuscript might be improved, and better aligned with the objectives of the journal which seeks 'research studies that are of high technical quality and are useful to the community' if there was less emphasis on blunt mathematical analysis and more attention was paid to sorting out epidemiological patterns or the ecology of the two diseases.

For example, although the authors state that evidence is lacking on whether maternal antibodies are passed on to young rabbits in the case of RHDV G1.2, it would be extremely surprising if it were not the case. This is a function of the rabbit immune system, not the virus. That could have been accepted as prior knowledge instead of it being 'confirmed' by the present study.

Reply: The authors agree that passive transfer of maternal immunity to juvenile rabbits was expectable (this is highlighted in the revised manuscript P5 L90-92), but respectfully disagree it should be considered prior knowledge. Passive transfer of maternal immunity to RHDV G1.2 has only been documented for experimentally vaccinated domestic rabbits, as mentioned in the Discussion (P9 L183-187). The present study provides the first evidence supporting maternal immunity upon natural infections in wild rabbits, although the alternative hypothesis that these patterns could be produced by early infections of very young rabbits cannot be ruled out with this data (discussed P9 L183-1871).

Looking at the graphs presented there are far more interesting and useful conclusions that could have been drawn from the data with a more specific analysis.

For example, Fig 1 (E) and Fig 2 (D) imply that there are very different patterns of outbreaks of RHDV and MYXV during the year with RHDV possibly having a bimodal pattern and MYXV being most prevalent in winter. However, without a proper analysis that corrects for the age of rabbits in the sampled populations in a more useful way it is not possible to sort out whether this might be due to vectors (e.g., rabbit flea abundance reaching a peak in winter) or interactions between the two viruses. For example, if RHDV killed many young rabbits that had not yet been exposed to the myxoma virus, the seroprevalence of MYXV antibodies could increase simply because the population would then include mostly older rabbits that had previously contracted the disease.

Reply: We believe that the seasonal patterns of the normalised absorption ratios to both viruses are not a simple reflection of viral circulation, but rather of the interaction between this and the breeding pulses that seasonally introduce in the population a large number of mostly seronegative rabbits. As a R-strategy species, rabbit populations are governed by their reproductive capacity and are thus extremely volatile in epidemiological terms. Therefore, in 2-3 months in the spring (and sometimes also in the autumn, depending on primary productivity), the population turns from comprising only adults, many of them seropositive, to being >80% juveniles, most of which are seronegative. This highly dynamic system precludes direct inference on viral circulation from semi-quantitative serological data and is discussed in the manuscript (P10 L206-216).

In a second example, the authors note that many young rabbits were seropositive for RHDV antibodies and assume these to be of maternal origin. The proportion of young rabbits with such antibodies declines until rabbits are about 63-64 days old then increases again, presumably because, on losing maternal antibody protection they have a higher chance of becoming infected with RHDV. However, there was no further analysis to support that evidence. Quite clearly, based on available data, only a proportion of the adult population was seropositive and showed evidence of prior exposure to RHDV, and so it would be expected that only a similar proportion of new-born young could show maternal antibodies. The broad statistical methods used do not appear to have been adequate to bring to the fore subtle inferences of this kind. Instead, careful sorting of the most relevant data (e.g., separating out seropositive young rabbits or seeking alternative ways of establishing the origin of antibodies)

would be necessary. Maternal antibodies are IgG isotype alone and iso-ELISA tests could have been used to confirm their origin.

Reply: We agree that isotype ELISAs could provide further evidence on the presumed maternal origin of IgG detected in juvenile rabbits. The use of ELISAs targeting IgM, which would identify recent infections, to answer this question is discussed in the manuscript (P9 L190-194). We are currently setting up such an ELISA to further study this research question.

Figure 3 is confused as the authors seek different ways of presenting their evidence that rabbit antibodies are boosted by re-exposure or reinfection with the disease. As it has been accepted for over 20 years that rabbit antibodies increase sharply on re-exposure, it would have been more useful to consider whether seropositive rabbits showing evidence of an antibody boost were more commonly detected in different seasons or months of the year. Such information would provide an independent assessment of the timing of RHDV outbreaks and help to interpret earlier presented data on the seasonal changes in antibody prevalence.

Reply: We acknowledge that Fig. 3 is hard to read due to sheer amount of information in the plots. We tried to simplify it by dividing the individual histories according to the number of recaptures, from more detailed histories with >4 captures (Fig. 3A) to the more incomplete ones with only 2 captures (Fig. 3C). We do not pretend this to be novel information, but to provide non-statistical evidence that seroconversions were common but no clear seroreversions were detected. These observations support the lifelong immunity to both viruses in free-ranging populations, which is indeed novel information.

Unfortunately, the interesting analysis of the seasonality of IgG boosts is not possible with this type of study, as data are not evenly distributed because we do not control the timing of recapture. As an example, the individual rabbit with a more detailed life history (9 captures) showed a two-fold increase in RHDV GI.2 normalised absorption ratios between consecutive captures, but these took place 9 months apart, precluding any conclusion on the seasonality of the presumed re-infection.

In short, although the authors consider that 'longitudinal serological studies play a key role' in understanding disease, I feel that they could have applied better analyses to the data available by initiating the manuscript with some specific hypotheses to be tested. This would have helped to focus the statistical analyses and helped in reaching more useful, or at least less speculative conclusions.

Reply: As mentioned in the Introduction, the aim of this study was to “... describe the patterns and assess the factors influencing the dynamics of antibodies generated upon a natural infection with MYXV and RHDV GI.2 in wild European rabbits.” (P5 L106-109). The results obtained in this study raise multiple hypothesis, which will be the subject of ongoing and future research.

We respectfully disagree with reviewer 1 that the statistical analysis could be more focused, as we performed linear mixed models controlling for the effect of many known covariates. The appropriateness of the statistical analysis is highlighted by the conditional R-squared values obtained (~90 %), meaning these models explain almost all the variance in the normalised absorption ratios of IgG to MYXV and RHDV GI.2.

The authors believe that the conclusions of the study will be useful to the community as they illustrate, as few other studies have done, how the long-term dynamics of humoral immunity can be addressed through longitudinal serological studies. Besides, the study provides evidence supporting, for the first time, the lifelong immunity to MYXV and RHDV GI.2 in wild populations of an endangered keystone species and the presence of maternal immunity to RHDV GI.2.

The mentioned speculative nature of the conclusions probably relates with the fact that this was an observational study. As such, it cannot prove causality, so the discussion of the results involves some degree of interpretation. Causality could be proven through experimental studies, but these are most difficult to perform when dealing with natural infections in wild populations.

Reviewer #2 (Comments for the Author):

The manuscript by Coelho et al presents the results of an extensive longitudinal field analysis on the development of protective anti viral immunity against important rabbit pathogens. The effort is remarkable and data presented indicates formation of lifelong protection after a few rounds of infection. While I liked the concept and the data, the presentation requires some rewording. In general, figures are hard to understand and pertinent information is not well conveyed. Please edit accordingly

Reply: We appreciate the comments from reviewer 2 and attempted to improve the presentation. Please see our response to the specific comments.

Specific comments:

Figures: What is the unit of iELISA result? It is not indicated in the legend.

Reply: We acknowledge the comment. The Normalised Absorbance Ratios are unitless, given they are computed as a ratio between the absorbance of the sample and the absorbance of the negative control. This is now made clear in the revised manuscript (Methods P15 L322-323). The Y-axis labels in the revised manuscript were corrected to “Normalised absorbance ratios”.

Figure 1B: What is the point of correlating with weight? Are these mice healthy and just bigger or overweight so obesity and immune system interplay is questioned?

Reply: We use body weight as a quantitative proxy of age, as these two variables are well correlated up to 800 g/4 months of age (Ferreira & Ferreira, 2014). Using this methodological approach is critical to address the question on the existence of maternal immunity in wild rabbits. In fact, using a categorical variable adult/juvenile (up to 800 g/4 months of age) would not highlight putative maternal antibodies that are expected to wane in the first 1-2 months of life. This is made clear in the revised manuscript (Methods, P13 L285-286, and P16 L340-341). The relationship between body condition (obesity) and humoral immunity is a very interesting research question. We intended to address this question by including body condition as a variable in the model, but decided not to as including this variable led to overfitting and poor convergence of the models.

Figure 3 is very hard to follow. Some other way of interpretation that includes how many times the rabbit is recaptured as a variable should be included.

Reply: The authors acknowledge the comment, as the large sample sizes makes it difficult to interpret Figure 3. In the revised manuscript we show the antibody dynamics separately according to the number of captures of each rabbit (≥ 4 , 3, or 2), to allow better visualization of the results that support the conclusion that humoral immunity against both viruses is likely lifelong.

Overall: Figure legends are too vague. More description is required for the readership to follow the complex data presented. For instance there is a dashed line in all graphs of but it is not acknowledged in any of the legends.

Reply: We acknowledge the comment and improved figure legends. The dashed line representing the cut-off threshold for seropositivity is now mentioned, as should have been from the start.

Discussion: "The role of the passive transfer of maternal immunity to juvenile rabbits, resulting in herd immunity with a low case-fatality rate, was shown for MYXV (33, 42). Our results suggest the same could occur with RHDV GI.2, but further analysis including specific IgM will be necessary to fully address this hypothesis." Why is IgM analysis required to test this hypothesis? IgM does not cross placenta. I do not get the relevance.

Reply: Our hypothesis is that IgG antibodies in juvenile rabbits might mean: 1) maternal immunity or 2) early infections. Testing for IgM in juvenile rabbits would signal recent infections, thus testing the second hypothesis. This is now made clearer in the revised manuscript (P9 L193-197): "Our results suggest the same could occur with RHDV GI.2, although early infections of juvenile rabbits could also produce the observed pattern. Further analysis including specific IgM to distinguish maternal immunity from recent infections will be necessary to fully address this hypothesis."

General: Did authors perform any studies on Ig subtypes? Do they predict any correlation with protection and certain isotypes?

Reply: The authors appreciate the interesting comment which could open a new line of research. We are currently setting up an isotype ELISA, specifically targeting IgM (please see our response to the previous question) but were not planning to address IgG subtypes.

In another study from our team, which is currently under review, we showed that free-ranging European rabbits with IgG normalised absorbance ratios at the cut-off threshold for positivity are fully protected from fatal infection during natural outbreaks of RHDV GI.2, while mortality was >75% in seronegative rabbits (Jiménez-Ruiz et al., under review). We are planning a similar study concerning Myxomatosis outbreaks. Isotyping and IgG subtyping seronegative rabbits that either died or survived outbreaks could provide evidence on the potentially differential protection provided by immunoglobulin isotypes and IgG subtypes upon natural infections.

Furthermore, IgG subtyping, particularly concerning IgG1 and IgG4, could help in addressing the role of re-infections in the long-term dynamics of humoral immunity to these viruses.

Reviewer #3 (Comments for the Author):

This is an extremely valuable paper for the researchers interested in RHDV and Myxoma, showing how monitoring of the animals in nature may contribute to expanding the biological features of the disease and the characteristics of the virus. The team deals with this kind of animal trapping for several years, I have no doubts that the study has been conducted properly. The paper is written with a very high consciousness and knowledge about the subject. The results are properly documented. The conclusion is very crucial, showing the nature of the infection and the humoral response to the infection in the natural conditions for the first time, especially in the means of Lagovirus europaeus Gl.2. I have no concerns about the paper and in my opinion it is worth publishing.

Reply: We appreciate the comments from Reviewer 3.

May 19, 2023

Dr. Nuno Santos
Associacao BIOPOLIS - Rede de Investigacao em Biodiversidade e Biologia Evolutiva
CIBIO
Campus de Vairao
Vairao 4485-661
Portugal

Re: Spectrum00050-23R1 (Dynamics of humoral immunity to myxoma and rabbit haemorrhagic disease viruses in wild European rabbits assessed by longitudinal semi-quantitative serology)

Dear Dr. Nuno Santos:

Your manuscript has been accepted, and I am forwarding it to the ASM Journals Department for publication. You will be notified when your proofs are ready to be viewed.

Sincerely,

Daniela Rajao
Editor, Microbiology Spectrum
